# Influences of study design on the effectiveness of consensus messaging: The case of medicinal cannabis

**Asheley R. Landrum**[1]*, **Brady Davis**[1], **Joanna Huxster**[2], **Heather Carrasco**[3]

1 College of Media & Communication, Texas Tech University, Lubbock, TX, United States of America,
2 Department of Environmental Studies, Eckerd College, St. Petersburg, FL, United States of America,
3 Rawls College of Business, Texas Tech University, Lubbock, TX, United States of America

* A.Landrum@ttu.edu

**Data Availability Statement:** Data and code can be found on our project page on OSF.io at https://osf.io/38rju/ (DOI: 10.17605/OSF.IO/38RJU).

## Abstract

This study examines to what extent study design decisions influence the perceived efficacy of consensus messaging, using medicinal cannabis as the context. We find that researchers' decisions about study design matter. A modified Solomon Group Design was used in which participants were either assigned to a group that had a pretest (within-subjects design) or a posttest only group (between-subjects design). Furthermore, participants were exposed to one of three messages—one of two consensus messages or a control message—attributed to the National Academies of Sciences, Engineering and Medicine. A consensus message describing a percent (97%) of agreeing scientists was more effective at shifting public attitudes than a consensus message citing substantial evidence, but this was only true in the between-subject comparisons. Participants tested before and after exposure to a message demonstrated pre-sensitization effects that undermined the goals of the messages. Our results identify these nuances to the effectiveness of scientific consensus messaging, while serving to reinforce the importance of study design.

## Introduction

The Gateway Belief Model (i.e., GBM [1]), argues that communicating about scientific consensus to the general public indirectly influences change in people's support for policies by first increasing their perceptions of scientific consensus and then aligning their attitudes with that of the scientists [1]. The vast majority of studies using the GBM examine consensus messaging in the context of climate change [2–4]. A handful of studies have also examined the GBM in the context of genetically modified organisms [5, 6], and at least one study so far has looked at the effects of consensus messaging on the issue of vaccination [7]. This study examines consensus messaging about the efficacy of cannabis for treating chronic pain.

According to the GBM, messages about scientific consensus on climate change correct faulty assumptions about the robustness of such consensus (measured as participants' estimates of the percent of scientists who agree with a proposition). These corrected beliefs then

**Funding:** The authors received no specific funding for this work.

**Competing interests:** The authors have declared that no competing interests exist.

influence individuals' views and attitudes about the risks posed by climate change, which influence support for relevant policies [4, 7].

The model has been tested primarily in the context of climate change, using the "97% of climate scientists agree. . ." message with a pie graph highlighting the 97% number. Concerns exist among some, however, regarding the applicability of these results outside of climate change. First, not all consensus messages can be accurately summarized as a proportion of scientists who agree (and arguably, consensus about climate change should not be interpreted that way either [8]). To this end, this study uses and compares two consensus messaging strategies. The first highlights the same numerical percentage that is used by the climate change GBM studies, 97%, here attributed to medical, as opposed to climate, scientists. Notably, studies that have used percentages lower than 97% to 98% have found weaker support [9, 10]. The proportion of medical scientists who believe that cannabis is an effective treatment for chronic pain has not been established in the way that proportions of agreeing scientists on other issues have [11] or the way that consensus estimates have been established on climate change [12], but we include this condition for the purpose of comparison. We call this strategy the descriptive norm/authority appeal as it takes the social norms approach to changing people's behavior by describing what others "think and do," but, instead of describing lay publics' social group members [13], the message describes views of scientists who are epistemic authorities. The second messaging strategy is accurate to the case of consensus surrounding medical cannabis use for chronic pain—a message we developed from a report written by a consensus panel formed by the National Academies of Sciences, Engineering, and Medicine (i.e., NASEM). One of the findings of this report is that there is substantial evidence that cannabis is an effective treatment for chronic pain in adult patients [14]. We call this the "evidence message" as it puts more emphasis on the weight of the evidence evaluated by a panel of experts as opposed to naming a proportion of agreeing scientists.

Although this "evidence message" design may be a closer description to what philosophers of science would label scientific consensus, and it may be more in line with how consensus is established [8], there is evidence that it may be a less effective communication strategy than the descriptive norm/authority appeal. For example, Myers et al. [3] found that when agreement among scientists was described but a numerical estimate was not used (e.g., "an overwhelming majority of scientists have concluded. . ." vs "97% of scientists have concluded"), participants' estimates of scientific consensus and other variables of interest did not significantly differ from the control condition. Similarly, Landrum et al. [6], which used a message highlighting a NASEM consensus panel on genetically modified organisms, also found no significant difference between exposure to the consensus message and participants' estimates of agreement among scientists. Landrum and Slater [8] propose that messages may be more or less successful depending on whether the question about estimating consensus is aligned with the message design. That is, if the message describes a proportion of agreeing scientists, the question asked to participants ought to be "what percent of scientists agree." On the other hand, if the message designed describes the process of consensus or a body of evidence, the question asked to participants ought to be to what extent they agree that consensus exists or that most of the evidence is supportive. To examine this, we randomly assigned participants in the current study to receive either the numerical ("what percent of scientists. . .") or the agreement ("to what extent do you agree that. . .") version of the consensus estimate question.

Another concern related to the applicability of the GBM outside of the climate studies relates to the choice of mediating variables used in the model. The GBM includes three mediating variables, two of which are specific to climate change: belief that climate change is *real* and belief that climate change is *caused by humans*. These first two mediating variables are expected to influence the third mediating variable, *worry* about climate change. Although the

**Table 1. Survey items.**

| Variable | Question Text | Scale |
|---|---|---|
| Believable[1] | This message is ________________. | 0 Not Believable to 100 Believable |
| Credible[1] | The source of this message, the National Academies of Sciences, Engineering, and Medicine, is _____________. | 0 Not Credible to 100 Very Credible |
| Deceptive[1] | The message is ________________. | 0 Not Deceptive to 100 Very Deceptive |
| **Perceptions of Consensus** | | |
| dns[2] | What percent of medical scientists do you believe agree that there is substantial evidence that marijuana/cannabis is effective for the treatment of chronic pain? | 0% to 100% |
| dnp[2] | What percent of the U.S. public do you believe agree that there is substantial evidence that marijuana/cannabis is effective for the treatment of chronic pain? | 0% to 100% |
| cns[3] | To what extent do you agree or disagree that there is consensus among the medical scientific community that marijuana/cannabis is effective for the treatment of chronic pain? | 0 Strongly Disagree to 100 Strongly Agree |
| cnp[3] | To what extent do you agree or disagree that there is consensus among the U.S. public that marijuana/cannabis is effective for the treatment of chronic pain? | 0 Strongly Disagree to 100 Strongly Agree |
| **Attitudes** | | |
| eff | To what extent do you, personally, believe that marijuana/cannabis is effective for the treatment of chronic pain? | 0 Not Effective to 100 Very Effective |
| safe | How safe do you, personally, believe using marijuana/cannabis is? | 0 Not at all safe to 100 Very safe |
| rmed | How much risk do you believe medical marijuana/cannabis poses to human health, safety, and/or prosperity? | 0 No risk at all to 100 Very high risk |
| rrec | How much risk do you believe recreational marijuana/cannabis poses to human health, safety, and/or prosperity? | 0 No risk at all to 100 Very high risk |
| **Policy Support** | | |
| ma21 | Medical marijuana/cannabis should be made legal for adults ages 21 and older | 0 Strongly disagree to 100 Strongly agree |
| mall | Medical marijuana/cannabis should be made legal for people of all ages, including those under 18. | 0 Strongly disagree to 100 Strongly agree |
| ra21 | Recreational marijuana/cannabis should be made legal for adults ages 21 and older | 0 Strongly disagree to 100 Strongly agree |
| rall | Recreational marijuana/cannabis should be made legal for people of all ages, including those under 18. | 0 Strongly disagree to 100 Strongly agree |

[1] Items asked only at time 2 (after being presented with the message).

[2] Half of the sample were asked to estimate percentage of agreement at time 2.

[3] Half of the sample were asked to what extent they agree or disagree that consensus exists at time 2.

other two items cannot, the *worry* item can be modified for other contexts to represent perceptions of risk about the issue at hand. In the case of cannabis, we asked participants how much risk they believe both medicinal and recreational cannabis pose to human health, safety, and/or prosperity. In addition to this risk perception question, we also asked participants how safe they feel using cannabis is and to what extent they personally believe that cannabis is effective for the treatment of chronic pain. See Table 1.

Furthermore, in many of the attempts to implement the GBM to test for potential indirect (and direct) effects of scientific consensus messaging, researchers have used only between-subjects manipulations [6, 15]. However, in the original and subsequent GBM papers by the original authors [1, 4], pre- and post-message exposure data is collected and the difference scores are used in the mediation model. Although it may not be immediately clear from visualizations of the GBM (e.g., Fig 1 [1]), condition (consensus message vs. control) is expected to predict *change* in perceived scientific agreement between time 1 and time 2, which is expected to predict *change* in beliefs (climate change is real, climate change is human caused) between time 1

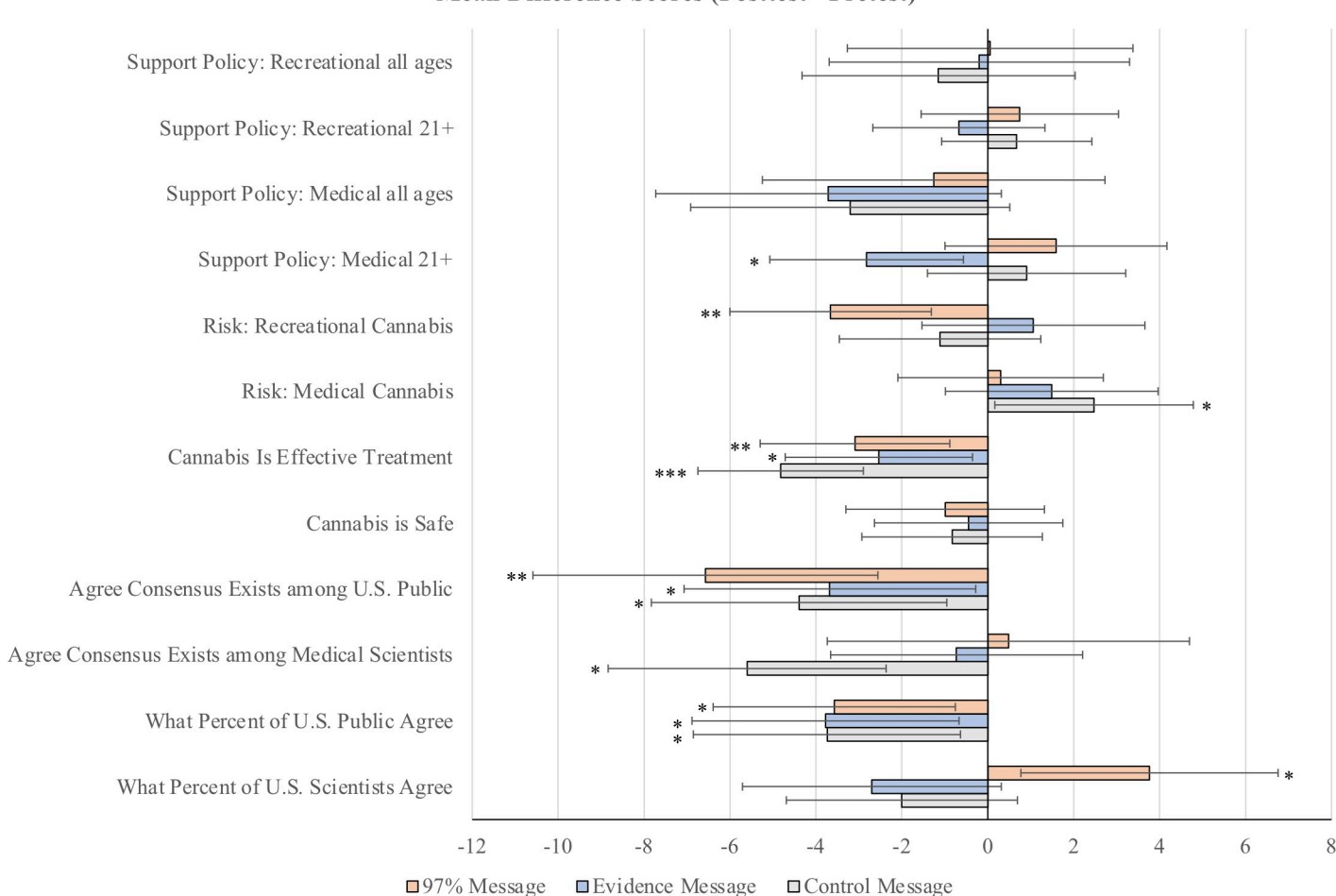

**Mean Difference Scores (Posttest - Pretest)**

**Fig 1. Mean difference scores by condition and question for the pretest/posttest sample.** Error bars represent 95% confidence intervals. There was approximately one week between pretest and posttest. ***$p < .001$, **$p < .01$, *$p < .05$ for two-tailed, single sample $t$ tests.

and time 2, etc. To be consistent with the original intention of the GBM and to test for differences between these two designs, we conducted a modified Solomon group design in which we collected both pretest/posttest data and posttest-only data.

## Current study

This study aims to contribute to our understanding of the efficacy of consensus messaging by examining how researchers' decisions about study design (e.g., whether data collected is cross sectional or pretest/posttest, how variables are operationalized, how consensus is approached and described, [8] influence study results. As stated earlier, we examine these questions using cannabis as the context. We chose medicinal cannabis as the context for a few reasons. First, scientific consensus has been established for this issue: a consensus panel convened by the National Academies of Science, Engineering, and Medicine (NASEM) determined that there is substantial evidence that cannabis is an effective treatment for chronic pain in adults [14]. Second, like for other issues for which consensus messaging has been studied (e.g., climate change, genetically modified organisms, vaccines), public policy arguably does not align with the available scientific evidence; despite its promising effects, medical cannabis remains illegal

in many states (17 at the time of data collection) and about one-third of the U.S. public oppose legalizing cannabis [16]. In fact, according to the consensus report, regulatory barriers—such as the classification of cannabis as a Schedule I substance—hinder the advancement of research on cannabis [14]. Using cannabis as an example, the current study tests and challenges aspects of the Gateway Belief Model, which provides an explanation for how scientific consensus messaging may improve public support for policies related to publicly controversial science.

## Methods

The study was approved by the Institutional Review Board at Texas Tech University as exempt research involving human subjects (IRB2020-302). Data were collected from a national sample of 1,558 U.S. adults recruited using Amazon's Cloud Research Services tool at the end of June 2020 and beginning of July 2020. Prior to answering any study questions, participants read a digital consent form that explained the study and the participants' rights and provided contact information for the IRB office and the principal investigator. Participants were then asked whether they consented to participate in the study. Participants who selected yes continued on and participants who said no were redirected to the end of the survey.

Participants ranged in age from 18 to 82 ($M$ = 41.11, $Median$ = 39, $SD$ = 13.28). For self-identified race and ethnicity, 9.3% of the sample reported identifying as Black or African American, 6.6% reported identifying as Hispanic/Latino, and 8.9% reported identifying as Asian; and 52.3% of the sample identified as female. The highest level of education earned for 8% of the sample was high school, around 31% of the sample completed at least some college coursework, and around 60% had at least a college education. Furthermore, 48.16% of the sample indicated that they were somewhat to very liberal, 22.56% were moderate, and 29.28% were somewhat to very conservative.

We conducted our survey experiment using a modified Solomon group design. This design is used to test for pretest sensitization, which occurs when participants' posttest ratings are influenced by exposure to pretest questions, but also to test for any differential condition effects based on whether participants completed the pretest [17]. We included a one-week gap between the pretest (time 1) and the experiment with posttest (time 2).

During the time 1 pretest, 956 participants (the pretest/posttest sample) were introduced to the topic of medical cannabis While we exclusively use the term cannabis in this manuscript, we intentionally deviate slightly in the experimental instrument. The term marijuana is commonly used within the context of legislature and regulatory guidelines (Romi et al., 2021). Therefore, in an effort towards ecological validity, we use the terms cannabis and marijuana interchangeably within the experimental instrument and any use of either terms appears as was presented to the participants with the following prompt which was adapted from content on the Mayo Clinic's website [18].

*Medical marijuana—also called medical cannabis—is a term for derivatives of the cannabis sativa plant that are thought to relieve serious and chronic symptoms. Some states allow marijuana use for medical purposes. Federal law regulating marijuana supersedes state laws. Because of this, people may still be arrested and charged with possession in states where marijuana for medical use is legal. In this study we will ask you about your views towards the use of both recreational and medical marijuana*

Following the prompt, participants were asked a series of questions about their perceptions of consensus surrounding the use of medical and recreational cannabis both from medical scientists and the U.S. public, their own attitudes towards recreational and medical cannabis, and

their support (or lack thereof) for legalization policies. Participants who completed the time 1 pretest survey were marked as eligible to sign up for the time 2 posttest survey a week later, and 935 returning participants completed this posttest survey. During the same time period, 610 new participants (posttest-only sample) completed the posttest-only survey at time 2, which was identical to the posttest survey. To ensure we would not have duplicate participants, those who took the pretest/posttest surveys were marked ineligible to sign up for the posttest-only survey and vice versa.

At time 2, participants (from both the pretest/posttest sample and posttest-only samples) were randomly assigned to one of three message conditions: (1) a consensus message stating that there is substantial evidence that cannabis is effective for the treatment of chronic pain in adults (i.e., evidence message), (2) a descriptive norm/authority consensus message stating that 97% of medical scientists agree that cannabis is effective for the treatment of chronic pain in adults (i.e., 97% message), and (3) a control message stating that researchers are investigating the potential uses of Cannabidiols (CBD), a compound in cannabis that does not have psychoactive effects (i.e., control message). All three messages were attributed to the National Academies (NASEM). These messages are available on our project site on osf.io (https://osf.io/w8u6k/). Following exposure to the message, as a manipulation check, we asked participants which of the following statements best describes the main point of the message they just saw: (a) There is scientific consensus that cannabis is effective for the treatment of chronic pain, (b) Research on the effectiveness of cannabis is still ongoing, (c) I don't know, or (d) I prefer not to answer. Overall, participants were accurate at identifying the main point of the message. Approximately 87% of the control participants chose option *b* (i.e., research is still ongoing), and 93% of the descriptive norm condition participants and 89% of the evidence message condition participants correctly chose option *a* (i.e., scientific consensus exists).

Furthermore, at time 2, we randomly assigned participants to answer one of two question formats about their perceptions of consensus. Half of the sample was asked to estimate what percent of medical scientists and what percent of the U.S. public agree that cannabis is effective for the treatment of chronic pain (on scales from 0 to 100%). The other half of the sample was asked to what extent they agree or disagree that there is consensus among medical scientists and among the U.S. public that cannabis is effective for the treatment of chronic pain (on scales of 0 –strongly disagree to 100 –strongly agree). Recall, during the pretest at time 1, participants were asked to answer both questions.

See our project page on the Open Science Framework https://osf.io/w8u6k/ for data files, R script, stimuli, and survey questions.

## Results

### Test for pretest sensitization effects on the condition manipulation

Following the recommendations of Braver and Braver [19] for analyzing Solomon group designs, we began by determining whether an interaction effect exists between our condition manipulation and the sample (pre/posttest sample, posttest-only sample). The presence of a significant interaction would indicate both that pre-sensitization exists *and* that it likely moderates any effects of the condition manipulation. Because we had several outcome variables, we first conducted a multivariate analysis of variance (MANOVA) Because we split the sample for the perceptions of consensus items, we left these four variables out of the MANOVA and found a significant main effect of our condition manipulation (Pillai's Trace = 0.053, approximate $F = 3.62$, $p < .001$). There was also a significant main effect of sample (Pillai's Trace = 0.021, approximate $F = 2.86$, $p = .001$), suggesting pre-test sensitization exists. Importantly, there was a significant interaction between sample and condition (Pillai's Trace = 0.024,

approximate $F = 1.64$, $p = .030$), meaning that the effects of our experimental manipulation are likely conditional on whether participants answered pretest questions.

Because we found evidence that pre-test sensitization exists and it likely affects the condition manipulation, we followed up on the significant main effect of condition using simple effects tests on the pre/posttest sample (within-subject effects) looking for a difference between time 1 and time 2 as well as simple effects of the condition manipulation on the posttest-only sample [19]. Descriptive statistics for the outcome variables for each sample are reported in the S1 Table. Full analyses results, including ANOVA tables, are available in the supplementary materials.

## Test for within-subject effects of condition manipulation—a difference score analysis

Our study design allowed us to examine whether the differences between individuals pre- and post-message exposure ratings (i.e., their "difference scores") varied significantly from 0—in other words, were there significant increases or decreases in the outcome variables between time 1 and time 2. We calculated difference scores by subtracting participants' pretest ratings *from* their posttest ratings (posttest—pretest = difference score) and then conducted single sample *t*-tests We conducted single-sample *t*-tests on the difference scores as opposed to paired-samples *t*-tests because the visual depiction (see Fig 2) is simpler—i.e., there are fewer bars to keep track of. We include the analysis using paired samples *t*-tests in the supplementary materials on OSF. Note that the results do not differ based on which version of *t*-test we use. Note that these analyses are only of the pretest/posttest sample. Fig 1 shows the mean differences by question and message condition.

Participants often moved in the non-expected direction between the pretest and posttest surveys. For instance, in all three conditions, participants said they agreed *less* that cannabis is effective for the treatment of chronic pain after they were exposed to the messages, even though two of the messages stated that there is consensus that cannabis is an effective treatment. Similarly, in all three conditions, participants expected a lower level of consensus among the U.S. public on the effectiveness of cannabis to treat chronic pain than before message exposure. Notably, the messages did not mention public views, only scientific ones. Though some may speculate that this move in the less desirable direction among participants could be boomerang or reactance effects [20, 21], an alternative explanation is that this is the result of pretest sensitization, especially given that this shift was seen in the control condition as well. In this case, discussing the issue incidentally may have made medicinal cannabis seem like a more controversial issue among the public than study participants originally thought.

We did find two significant expected effects for the condition in which participants were exposed to the 97% message: participants in this condition rated recreational cannabis as less risky than they did at time 1 (although they didn't shift on medical cannabis which is what the message was about), and they increased the percent of scientists presumed to agree from time 1 to time 2. See Fig 1 and S2 Table.

## Test for between-subject effects of condition manipulation

Because we initially found a significant interaction between our condition manipulation and the sample from the MANOVA, which indicates that pre-sensitization exists and that it likely moderates any effects of the condition manipulation, we analyzed our pretest/posttest sample and posttest-only sample separately [19]. To analyze our posttest sample, we followed up on the MANOVA with one-way ANOVAs on each of the dependent variables (see Table 1). In this case, we were specifically looking for differences between the consensus message

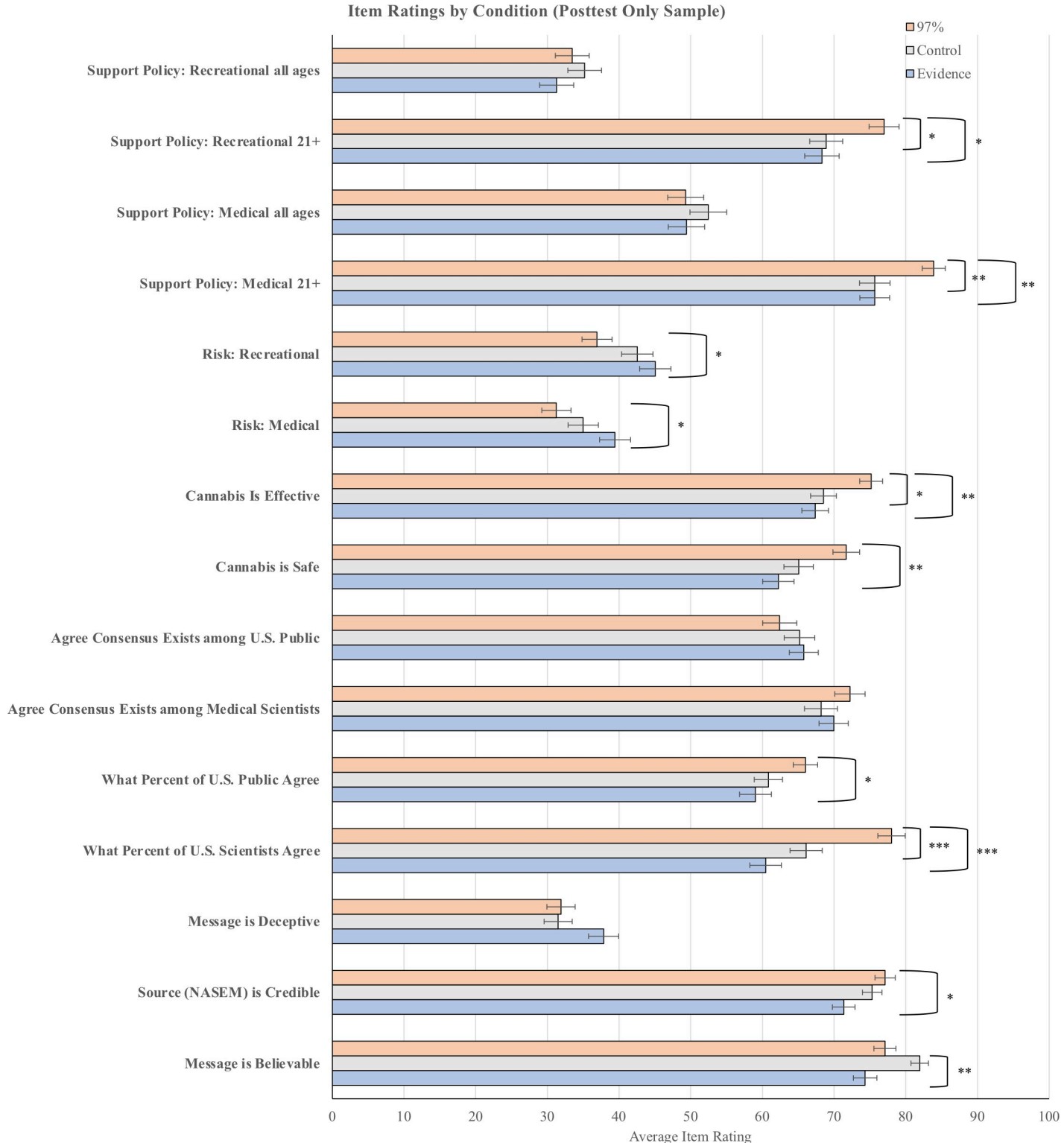

**Fig 2. Mean rating for each item by condition and question for the posttest only sample.** Error bars represent standard error. Significant differences between message conditions (determined by Tukey tests) are shown. ***p < .001, **p < .01, *p < .05.

conditions (the 97% message, the evidence message) and the control message. See Fig 2 and S3 Table.

Unlike the results from the pretest/posttest sample, the results from the posttest-only sample are more supportive of the hypothesis that there *are* effects of consensus messaging—at least when it comes to messages that describe a descriptive norm amongst authorities (i.e., 97% of medical scientists agree). Indeed, there were significant differences between the 97% message and the control message for four key variables: percent of scientists perceived to agree, belief that cannabis is an effective treatment for chronic pain, and support for legalizing medical and recreational cannabis for those aged 21 years and older.

Notably, these differences between the control condition and consensus message condition were not significant when the consensus message described substantial evidence (as opposed to a proportion of agreeing scientists). And in many cases, participants' item ratings in the 97% message condition differed significantly from those in the evidence message condition. This leads to questions about whether participants are actually influenced by the *consensus* aspect of the message or some other characteristic that differs between the two types of consensus messages examined in this study (e.g., the presence of a number, a high percentage).

## Aligning the message strategy with the measurement design

Sometimes the relationship between exposure to a consensus message and participants' estimates of scientific consensus is not statistically significant when non-numeric messages are used [3, 8, 22]. Landrum and Slater [8] hypothesize that this may be due to a lack of alignment between the type of message used and the way the participants' perceptions of scientific consensus is measured. For example, Bolsen and Druckman [23] found a significant relationship between exposure to a process-based consensus message (describing how consensus was formed from a National Academies of Sciences panel) and their measure of perception of scientific consensus (i.e., whether most scientists agree). We designed this study to answer this question by randomly assigning participants at posttest to two different forms of measurement for this question: one that asked them to estimate the percent of scientists who agree and one that asked them how much they agreed or disagreed that there is scientific consensus (see Table 1). We found mixed evidence regarding this hypothesis. See Table 2.

**Change in perceptions of scientific consensus between time 1 and time 2.** First, we looked at change in perceptions of scientific consensus among the pretest/posttest sample. A 3 (Message Condition) by 2 (Measure) ANOVA suggests that there is a main effect of condition, $F(2, 882) = 6.88$, $p = .001$, $\eta_p^2 = 0.02$, but not an effect of Measure, $F(1, 882) = 1.54$, $p = .215$, $\eta_p^2 = 0.002$, or an interaction effect between condition and measure, $F(2, 882) = 1.87$, $p = .155$, $\eta_p^2 = 0.004$. Follow-up simple GLM analyses show that the relationships between condition

**Table 2. Is there a significant relationship between condition manipulation and participants' perception of consensus based on consensus message strategy used and measurement?**

|  | Significant relationship between consensus message used and perception of consensus? | |
| --- | --- | --- |
|  | 97% vs. Control | Evidence vs. Control |
| *Pretest/posttest sample* |  |  |
| Δ Estimated percent of scientists who agree | Yes | No |
| Δ Agreement that consensus exists | Yes | Yes |
| *Posttest-only sample* |  |  |
| Estimated percent of scientists who agree | Yes | No |
| Agreement that consensus exists | No | No |

(consensus vs. control) and participants' perception of consensus are as follows. The relationship between condition manipulation—the 97% message compared to the control—significantly predicts both participants' estimates of the percentage of scientists who agree ($b$ = 5.76, $p$ = .005) and participants agreement that consensus exists ($b$ = 6.08, $p$ = .022). However, the relationship between condition manipulation—the evidence message compared to the control—predicts only participants' agreement that consensus exists ($b$ = 4.88, $p$ = .028) and not participants' estimates of the percentage of scientists who agree ($b$ = -0.70, $p$ = .732). This lends some support to the hypotheses that the measurement must be aligned with the message, but this seems to be true only for the evidence message.

**Perceptions of scientific consensus between conditions.**   Next, we looked at participants perceptions of scientific consensus at time 2 among the posttest-only sample. Unlike for the within-subjects data, a 3 (Message Condition) by 2 (Measure) ANOVA shows a significant interaction between message condition and measure, $F(2, 604)$ = 6.65, p = .001, $\eta p2$ = 0.02, in addition to the main effect of condition, $F(2, 604)$ = 12.64, p < .001, $\eta p2$ = 0.04. There was no significant main effect of measure, $F(1, 604)$ = 1.27, p = .260, $\eta p2$ = 0.002. Follow-up simple GLM analyses show the relationships between condition manipulation (consensus vs. control) and participants' perceptions of consensus are as follows. The relationship between condition manipulation—the 97% message compared to the control—significantly predicts participants' estimates of the percentage of scientists who agree (b = 11.91, p < .001) but not participants agreement that consensus exists (b = 4.04, p = .169). However, the relationship between condition manipulation—the evidence message compared to the control—does not significantly predict participants' agreement that consensus exists (b = 1.77, p = .540) nor participants' estimates of the percentage of scientists who agree (b = -5.65, p = .074). In this case, the alignment appears to matter for the 97% message, but not for the evidence message.

## Conceptually replicating the Gateway Belief Model

To test the hypothesis by the GBM that consensus messaging indirectly influences policy support by correcting people's estimates of scientific consensus and shifting their attitudes, we conducted mediation analysis using PROCESS (model 6 [24]). As stated earlier, the mediators in the GBM must be altered for different topics. The GBM for climate change includes three mediators after the estimate of the proportion of agreeing scientists—i.e., belief in climate change, belief in human causation, worry about climate change, two of which are not applicable to other issues like genetically modified organisms or cannabis. We used participants' perceptions of risk for medical cannabis. An alternative to this model using our data could use "effectiveness" in place of the risk perception. However, as the change in effectiveness was negative for each of the conditions, this didn't make sense to test for the current study See Fig 3. Furthermore, consistent with van der Linden et al. [1], each of the mediators and the outcome variable are difference scores (time 2 –time 1). For the model shown, the only effect we were able to replicate was the effect of message condition on the change in the estimated percent of agreeing scientists. No other paths (direct or indirect) were significant. For the full results, see the supplementary materials.

We also ran the model using the posttest-only data. Importantly, this model describes the relationship between the variables measured at time 2 and not the relationship between change scores the way that the original GBM specified. In the case of the posttest-only data, the model worked as predicted. In terms of direct effects, compared to the control condition, the descriptive norm condition is related to a greater proportion of medical scientists assumed to agree, which is related to lower risk perceptions for medical cannabis, which is then related to greater support for legalization policies. See Fig 4. Furthermore, the indirect effect of condition on

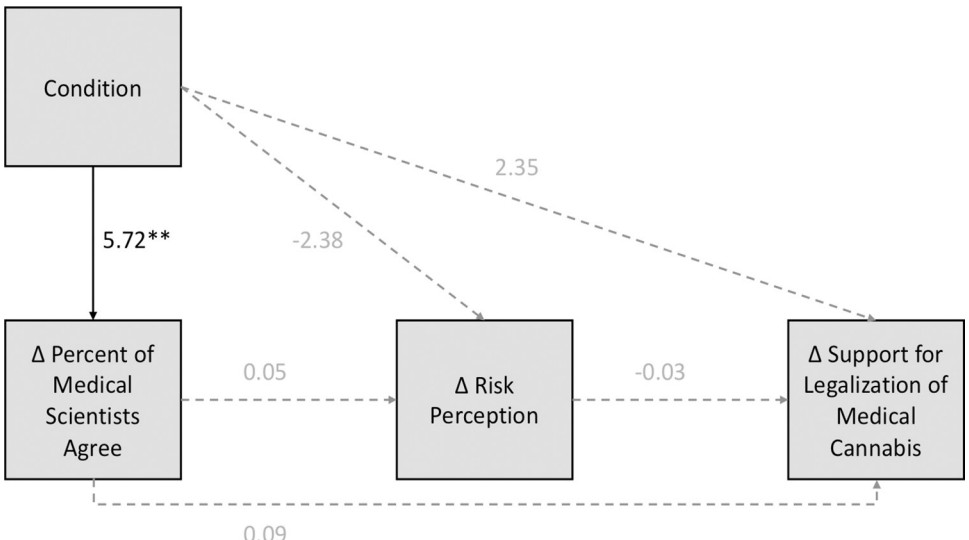

**Fig 3. Modified version of the GBM for medical cannabis using pretest/posttest data (change scores).** All shown paths were tested but the only significant path was from the message manipulation to the change in estimated percent of agreeing scientists. Note that condition only reflects the descriptive norm/authority message versus the control message and does not include the evidence message.

support for legalization of medical cannabis through perceptions of the percent of medical scientists who agree and risk perceptions was significant (b = 1.44, 95% CI [0.37, 3.06]).

## Discussion

This study aimed to contribute to our understanding of the efficacy of consensus messaging by examining how researchers' decisions about study design might influence study results, using medicinal cannabis as the context. We began by first testing for direct experimental effects on each of the outcome variables, including participants' beliefs about and estimations of

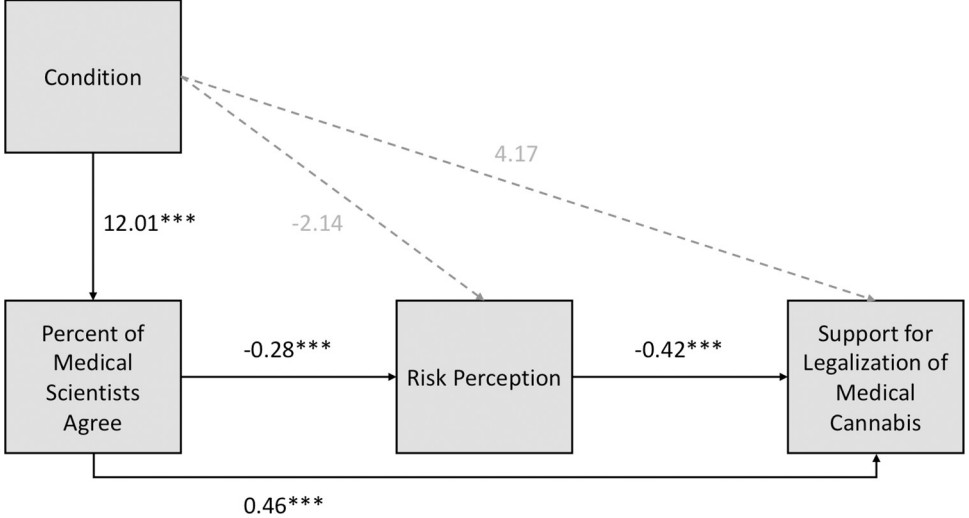

**Fig 4. Modified version of the GBM for medical cannabis using the posttest-only data.** Note that condition only reflects the descriptive norm/authority message versus the control message and does not include the evidence message.

consensus, beliefs about cannabis's efficacy for treating chronic pain, perceptions of risk associated with using medicinal and recreational cannabis, and support for policies legalizing their use. Then, we examined how researchers' decisions about study design (e.g., whether data collected is cross sectional or pretest/posttest, how variables are operationalized, how consensus is approached and described [8]) influenced study results. Finally, we tested two models aiming to conceptually replicate the GBM for determining whether the predicted indirect path from consensus messaging to policy support is present.

## Experimental effects of consensus messaging

First, we aimed to test for direct experimental effects on each of the outcome variables. Because of our study design, we were able to test this in two ways: whether participants changed their ratings of the outcome variables after being exposed to the messages (i.e., difference score analysis) and whether participants who were exposed to the consensus messages (as opposed to the control messages) had different ratings of the outcome variables (i.e., between conditions analysis). Notably, we found evidence of pretest sensitization and evidence suggesting that pretest sensitization influenced the effect of the condition manipulation. Therefore, we needed to analyze the two samples separately [19].

**Differences in ratings before and after exposure to the consensus message.** These specific consensus messages about the effectiveness of medical cannabis to treat chronic pain should have influenced participants to increase their perceptions of scientific consensus, potentially decrease their perceptions of risk associated with medical cannabis, increase their beliefs that medical cannabis is effective for the treatment of chronic pain, and potentially increase support for the legalization policies. However, we found that in many cases, participants moved in the non-hypothesized direction. For example, in all three conditions, participants shifted their ratings to agree less that medical cannabis is an effective treatment for chronic pain. Although pretest sensitization often assumes that the pretest will increase participants' awareness and/or responsiveness to the condition manipulation [19], here, participants often shifted in the opposite direction. One possibility for this, as we discussed earlier, is that participants may have begun to wonder if cannabis is a more controversial issue than they originally thought because we asked them these questions on more than one occasion.

**Differences in ratings between conditions for posttest-only sample.** Amongst the posttest-only data, we generally found that the 97% message appears to influence participants in the expected ways relative to the control. That is, compared to the control condition, participants in the 97% message condition estimated a larger proportion of agreeing scientists on average, were more likely to agree that cannabis is an effective treatment for chronic pain, and were more likely to support legalization for both medical and recreational cannabis for adults (ages 21 and older). Although this was the case for the 97% message compared to the control, this was not true for the "evidence" message compared to the control. We discuss the implications of this difference further below.

## Effects of study design decisions

The second aim of this research was to contribute to theory by examining how researchers' decisions about study design in the consensus messaging literature influence study results. We already discussed the differences associated with pretest/posttest data compared to cross-sectional data. We now discuss some of the other design decisions that appeared to influence the results.

**The descriptive norms/authority approach (97% message) appeared to be more influential than the description of the weight of evidence.** In the introduction, we discussed two

approaches to describing scientific consensus that we would test in this study: the descriptive norms/authority approach (i.e., the 97% message) and the evidence-based approach (i.e., the evidence message). We mentioned that prior work suggests that the descriptive norms/authority approach may be a more effective strategy for persuasion even if it is a less accurate representation of what scientific consensus is, and these previous findings were supported by our results. Even when the descriptive norm/authority message—the 97% message—didn't vary significantly from the control condition, it often varied significantly from the evidence message condition.

**Consensus messages influenced numerical estimates of consensus but not agreement that consensus exists.** One hypothesis from Landrum and Slater [8] was that the reason non-numeric messages (e.g., messages that stress the evidence or describe agreement without specific numbers) may not predict perceptions of consensus is that the question is often not aligned with the message and needs to be so for the treatment to be effective. That is, if a non-numeric question is asked, then the participants need to be asked to express a non-numeric form of the perception of scientific consensus (e.g., to what extent people agree that scientific consensus exists) rather than to estimate a numeric proportion of scientists in agreement. However, we found mixed evidence regarding this hypothesis. See Table 2. Future research should continue to investigate this.

## Conceptually replicating the Gateway Belief Model

Finally, we aimed to test a conceptual replication of the GBM for determining whether the predicted indirect path from consensus messaging to policy support is present. Interestingly, we failed to replicate the GBM when we constructed the model using difference scores as is consistent with the original GBM studies [1, 4]. However, we did find the expected relationships between variables when we used cross-sectional data. Using cross-sectional data predicts relationships between the ratings at time 2 as opposed to predicting the change in participants' ratings between time 1 and time 2 (i.e., difference scores). One reason that the first model (using difference scores) didn't show the hypothesized relationships may be related to the presence of the pre-sensitization effects and the conditional effects of our experimental manipulation based on the pre-sensitization.

## Limitations

Several limitations to this study need to be taken into consideration when interpreting the results. First, we collected the data via Amazon's Cloud Research platform. Although care was taken to attempt to get a diverse sample by requesting groups of participants based on age and religiosity, the sample is not nationally representative. This panel is generally known for having a more politically liberal panel (especially amongst older participants [25]), and this is true of our sample. However, although there are some differences between the political ideology groups in support of legalization policies, the strongest demographic predictor of support for cannabis legalization that has been reported is generation [16]. Differences in support for cannabis legalization based on gender, race/ethnicity, or education have not been found by prior nationally representative surveys [16].

A second limitation is that the data for this study was collected during the summer 2020 when many individuals were quarantining due to the COVID-19 pandemic. Some mainstream news coverage during the time of data collection suggested that cannabis may provide treatment for some of the side effects of COVID-19 [26], which could have increased support further for cannabis legalization. Indeed, cannabis sales escalated across the U.S. and Canada during this period [27, 28]; and in some states, cannabis dispensaries were considered

"essential businesses" [29]. However, other news coverage suggested that bills for cannabis legalization were being sidelined while local and state governments were focused on the pandemic [30].

A third limitation is that our control condition may not have functioned as we had intended. We chose to make the control condition about ongoing research related to CBD because we wanted a control message that was tangentially related to cannabis but was not a consensus message and was not about medical uses of cannabis. We expected that there would be no change between pretest and posttest for this control condition (e.g., CBD). According to the FDA, marijuana is different from CBD [31]. CBD is one compound in the cannabis plant, is not psychoactive (*c.f.*, tetrahydrocannabinol, or THC), and is marketed in an array of health and wellness products in places where cannabis remains illegal [31]. Most participants (87%) who were randomly assigned the control condition message about CBD understood that this message was NOT stating that scientific consensus exists, and they chose the option that indicated that research on the effectiveness of *cannabis* is still ongoing. Although the control condition message mentioned that "researchers are still investigating. . ." the topic of the investigation was CBD, and the participants may not have made a distinction between the two. In retrospect, we should have included a response option that specifically mentioned CBD and not cannabis. Importantly, though, we would still expect the message to work in a similar way as if it were clearly not about cannabis. Since participants generally seemed to understand that the message meant no consensus exists (because research is still ongoing), we would not expect change between pretest and posttest on our outcome variables. We would have only expected negative change for the control condition if the message had stated that there is scientific consensus that cannabis is NOT effective or if there was pretest sensitization. We have no reason to believe our results were due to the former as the manipulation check item suggests participants understood the purpose of the message. Thus, we do not believe our results were negatively affected by this potential issue.

Lastly, it is also worthwhile to consider that many of the consensus messaging studies have focused on issues for which public support is much lower than it is for cannabis. According to the Pew Research Center, in 2019, approximately only 8% of U.S. adults believe that cannabis should be kept illegal in all circumstances (medical and recreational); in contrast, 59% say it should be legal in all circumstances and 32% say it should be legal only for medical use [16]. Overall, in our data, support for medical cannabis was very high. At pretest, before seeing any consensus message, participants assumed an average of 72% of scientists ($SD = 18.19$, median = 75%) agreed that cannabis is effective for the treatment of chronic pain. They strongly agreed, themselves, that cannabis is an effective treatment ($M = 77.25$ out of 100, $SD = 21.17$, median = 81). They agreed that cannabis is safe ($M = 68.23$ out of 100, $SD = 28.3$, median = 75). And they perceived the risk of medical cannabis to be low ($M = 31.61$ out of 100, $SD = 29.16$, median = 20). Furthermore, support for legalization of medical cannabis for adults (21 and older) was high ($M = 80.73$ of 100, $SD = 26.51$, median = 91). Thus, there may be no need to use consensus campaigns to increase public support on this issue. We did see less support for legalizing medical cannabis for people of all ages, including children ($M = 53.48$ of 100, $SD = 36.38$, median = 60). So, future research could consider testing messages specifically discussing the effectiveness of medical cannabis among younger populations.

## Conclusions

This study provides more evidence that study design decisions influence the extent to which exposure to a consensus message influences public perceptions and indirectly influences policy support (as posed by the Gateway Belief Model). One such decision is the way in which

consensus messages are described; our study adds to the literature suggesting that the descriptive norm/authority appeal strategy is more persuasive than describing the existence of substantial evidence. However, as Landrum and Slater [8] discussed, there are philosophical issues with treating the descriptive norm/authority appeal strategy as a "consensus" message as well as practical issues (e.g., there is not always an accurate measurement of the proportion of agreeing scientists).

One obvious question might now be: what does the public know or think about scientific consensus? While less of an issue when using a specific percentage of scientists, as in the "97%" treatments for climate change, messages simply stating that there is a scientific consensus on an issue rely on the public understanding what constitutes such a consensus. On the contrary, the topic of scientific consensus appears to not be broadly understood. It may be that the oversimplification of consensus messaging overlooks the complexities which differentiate consensus from mere agreement. If this is true, a connection may exist between this vagueness in definition and public belief that consensus is manufactured and a product of group think [32, 33]. Further, sufficient understanding of the epistemic significance of the term "consensus" might not be attainable simply through learning a definition of the term [32]. In a large interview study, Slater et al. [33] find that few members of the general public are aware of the concept of scientific consensus at all, and that those who are familiar have a limited and unsophisticated grasp of it. This brings us to another apparent dilemma for consensus-framed science communication and particularly the use of the GBM for communicating science about which a percentage consensus message is not available: the public's limited understanding of the subject is likely to make messaging around it ineffective.

## Supporting information

**S1 Table. DFescriptive statistics for each of the outcome variables for the pretest/posttest sample and the posttest only sample.**
(DOCX)

**S2 Table. Descriptives, *p* values, and Cohen's *d* for single-sample *t*-tests (pretest/posttest sample).**
(DOCX)

**S3 Table. Between conditions effects for posttest only sample.**
(DOCX)

## Author Contributions

**Conceptualization:** Asheley R. Landrum, Brady Davis, Joanna Huxster.

**Data curation:** Asheley R. Landrum.

**Formal analysis:** Asheley R. Landrum.

**Methodology:** Asheley R. Landrum, Joanna Huxster.

**Project administration:** Asheley R. Landrum.

**Supervision:** Asheley R. Landrum.

**Visualization:** Asheley R. Landrum.

**Writing – original draft:** Asheley R. Landrum, Brady Davis.

**Writing – review & editing:** Asheley R. Landrum, Joanna Huxster, Heather Carrasco.

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
