## [Decision Letter · Decision Letter 0]

12 Oct 2021

PONE-D-21-22069Influences of Study Design on the Effectiveness of Consensus Messaging: The Case of Medicinal CannabisPLOS ONE

Dear Dr. Landrum,

Thank you for submitting your manuscript to PLOS ONE. After careful consideration, we feel that it has merit but does not fully meet PLOS ONE’s publication criteria as it currently stands. Therefore, we invite you to submit a revised version of the manuscript that addresses the points raised during the review process.

We look forward to receiving your revised manuscript.

Kind regards,

Lucy J Troup, Ph.D

Academic Editor

PLOS ONE

Journal Requirements:

2. We note that Figure 1 in your submission contain copyrighted images. All PLOS content is published under the Creative Commons Attribution License (CC BY 4.0), which means that the manuscript, images, and Supporting Information files will be freely available online, and any third party is permitted to access, download, copy, distribute, and use these materials in any way, even commercially, with proper attribution. For more information, see our copyright guidelines: http://journals.plos.org/plosone/s/licenses-and-copyright.

Additional Editor Comments (if provided):

Thank you so much for your patience. I would like to concur with the reviewers comments that this is a well written manuscript. It could however benefit from some minor revisions. Please look careful at the reviewers suggestions and I look forward to receiving your revised manuscript.

Reviewers' comments:

Reviewer's Responses to Questions

**Comments to the Author**

1. Is the manuscript technically sound, and do the data support the conclusions?

Reviewer #1: Yes

Reviewer #2: Yes

2. Has the statistical analysis been performed appropriately and rigorously? 

Reviewer #1: Yes

Reviewer #2: Yes

3. Have the authors made all data underlying the findings in their manuscript fully available?

Reviewer #1: Yes

Reviewer #2: Yes

4. Is the manuscript presented in an intelligible fashion and written in standard English?

Reviewer #1: Yes

Reviewer #2: Yes

5. Review Comments to the Author

Reviewer #1: This was a well written and organized manuscript that explored the effectiveness of consensus and evidence based messaging in view points on medical cannabis. Overall the authors found that a message reporting a high percentage of scientists agreeing is a more convincing than reporting existence of evidence. The researchers also found exposure effects related to research method. This is an important article as it can help media and researchers effectively get their message out and for researchers to choose the best method of testing method effectiveness.

It was not clear in the introduction as to why medical cannabis was the topic of choice. The authors noted that "we are less concerned with increasing public support for medical cannabis than we are curious about the persuasiveness of different messaging strategies." which is fine, but this manuscript will attract readers that are concerned with this, therefore I think a bit more needs to be discussed on the topic.

My major concern with the manuscript was the control condition. In Figure 1 the two experimental conditions state the "cannabis is effective" while the control condition mentions CBD and the fact that is has no psychoactive effects. I am not sure if that is a good control condition to compare against the experimental conditions, since it is only mentioning CBD when the questions asked were about cannabis. What I would like to see is a discussion on how this issue may have affected the results.

Minor things

The resolution for the figure was to poor to read them comfortably.

In the conclusion, please add a statement on study design your findings.

Reviewer #2: Development of concept using different messaging strategies is very interesting, well explained and referenced. Excellent range of testing used, interesting and statistically relevant outcome reported. Some very interesting findings in regards to a shift in perception post consensus messaging, and this would benefit from further exploration in other research papers. This paper has implications beyond the cannabis field and is an excellent contribution to the topic.

One point that needs work on:

Overarching research question at the start of the paper is whether consensus message or evidence messaging influenced the perception of medical cannabis. Throughout the paper it is clear there are other aspects being studies, however, in the discussion it states the overarching question for the study was whether consensus messaging influences public support for legalization of cannabis. This is not made clear at the start of the paper, and indeed there are several bits throughout the paper that set out how the study is being used to test different aspect of cannabis, and to see what impact the study design has consensus. This needs to be tidies up a bit and the focus of the paper clarified at the start/discussion.

6. PLOS authors have the option to publish the peer review history of their article (what does this mean?). If published, this will include your full peer review and any attached files.

Reviewer #1: No

Reviewer #2: **Yes: **Anna Ross

---

## [Author Response · Author response to Decision Letter 0]

21 Oct 2021

**NOTE: I have also uploaded this information as a Response to Reviewers document**

Authors’ Responses to Reviewers

Thank you for giving us the opportunity to revise our manuscript titled “Influences of Study Design on the Effectiveness of Consensus Messaging: The Case of Medical Cannabis” and resubmit for possible publication at PLOS One.

Below, we include point-by-point responses to the comments provided by the reviewers. In addition to the changes that we describe below, we have also removed Figure 1. Although we created the images/messages and used open access stock photos, we were not able to secure the rights to publish the figure with the logo from the National Academies of Sciences, Engineering, and Medicine. Therefore, we cut the figure from the paper and direct readers to our project page on OSF.io to view the stimuli used in the study.

Review Comments to the Author

**Reviewer #1: This was a well written and organized manuscript that explored the effectiveness of consensus and evidence-based messaging in viewpoints on medical cannabis. Overall, the authors found that a message reporting a high percentage of scientists agreeing is a more convincing than reporting existence of evidence. The researchers also found exposure effects related to research method. This is an important article as it can help media and researchers effectively get their message out and for researchers to choose the best method of testing method effectiveness.**

Thank you!

**REV: It was not clear in the introduction as to why medical cannabis was the topic of choice. The authors noted that "we are less concerned with increasing public support for medical cannabis than we are curious about the persuasiveness of different messaging strategies." which is fine, but this manuscript will attract readers that are concerned with this, therefore I think a bit more needs to be discussed on the topic.**

Our reason for emphasizing this is that we wanted to be clear that this is a manuscript that contributes to theory by examining the effectiveness of consensus messaging as a strategy for increasing public support for controversial policies (such as the legalization of cannabis). To address the reviewer’s concern, we have updated the “current study” paragraph by providing the reasons for which cannabis was chosen as the context for the paper. First, scientific consensus has been established for this issue, and second, the current policies in place do not align with the available scientific evidence. We also reworded the last line in this paragraph so that it does not downplay the context. It now states: “Using cannabis as an example, this research tests and challenges aspects of the Gateway Belief Model, which provides an explanation for how scientific consensus messaging may improve public support for policies related to publicly controversial science.”

**REV: My major concern with the manuscript was the control condition. In Figure 1 the two experimental conditions state the "cannabis is effective" while the control condition mentions CBD and the fact that is has no psychoactive effects. I am not sure if that is a good control condition to compare against the experimental conditions, since it is only mentioning CBD when the questions asked were about cannabis. What I would like to see is a discussion on how this issue may have affected the results.**

We added this to the limitations section of the paper. See this paragraph below:

A third limitation is that our control condition may not have functioned as we had intended. We chose to make the control condition about ongoing research related to CBD because we wanted a control message that was tangentially related to cannabis but was not a consensus message and was not about medical uses of cannabis. We expected that there would be no change between pretest and posttest for this control condition (e.g., CBD). According to the FDA, marijuana is different from CBD (FDA, 2020). CBD is one compound in the cannabis plant, is not psychoactive (c.f., tetrahydrocannabinol, or THC), and is marketed in an array of health and wellness products in places where cannabis remains illegal (FDA, 2020). Most participants (87%) who were randomly assigned the control condition message about CBD understood that this message was NOT stating that scientific consensus exists, and they chose the option that indicated that research on the effectiveness of cannabis is still ongoing. Although the control condition message mentioned that “researchers are still investigating…” the topic of the investigation was CBD, and the participants may not have made a distinction between the two. In retrospect, we should have included a response option that specifically mentioned CBD and not cannabis. Importantly, though, we would still expect the message to work in a similar way as if it were clearly not about cannabis. Since participants generally seemed to understand that the message meant no consensus exists (because research is still ongoing), we would not expect change between pretest and posttest on our outcome variables. We would have only expected negative change for the control condition if the message had stated that there is scientific consensus that cannabis is NOT effective or if there was pretest sensitization. We have no reason to believe our results were due to the former as the manipulation check item suggests participants understood the purpose of the message. Thus, we do not believe our results were negatively affected by this potential issue. 

**REV: Minor things**

**REV: The resolution for the figure was too poor to read them comfortably.**

We converted our original images from pdf to tiff (300 dpi) as requested by PLOS One’s image submission guidelines. We agree that the images as they appear in the submission pdf are blurry. However, if you click the “Click here to access/download; Figure…” the downloaded versions of the figures are much clearer. 

**REV: In the conclusion, please add a statement on study design your findings.**

We changed the first paragraph of the conclusions section to the following:

This study provides more evidence that study design decisions influence the extent to which exposure to a consensus message influences public perceptions and indirectly influences policy support (as posed by the Gateway Belief Model). One such decision is the way in which consensus messages are described; our study adds to the literature suggesting that the descriptive norm/authority appeal strategy is more persuasive than describing the existence of substantial evidence. However, as Landrum and Slater discussed (2020), there are philosophical issues with treating the descriptive norm/authority appeal strategy as a “consensus” message as well as practical issues (e.g., there is not always an accurate measurement of the proportion of agreeing scientists). 

**Reviewer #2: Development of concept using different messaging strategies is very interesting, well explained and referenced. Excellent range of testing used, interesting and statistically relevant outcome reported. Some very interesting findings in regard to a shift in perception post consensus messaging, and this would benefit from further exploration in other research papers. This paper has implications beyond the cannabis field and is an excellent contribution to the topic.**

Thank you!

**REV: One point that needs work on:**

**REV: Overarching research question at the start of the paper is whether consensus message or evidence messaging influenced the perception of medical cannabis. Throughout the paper it is clear there are other aspects being studies, however, in the discussion it states the overarching question for the study was whether consensus messaging influences public support for legalization of cannabis. This is not made clear at the start of the paper, and indeed there are several bits throughout the paper that set out how the study is being used to test different aspect of cannabis, and to see what impact the study design has consensus. This needs to be tidies up a bit and the focus of the paper clarified at the start/discussion.**

Thank you for this comment. First, we changed the first line of the discussion to state “This study aimed to contribute to our understanding of the efficacy of consensus messaging by examining how researchers’ decisions about study design might influence study results, using medicinal cannabis as the context.”

Furthermore, we realize that we were using “acceptance” and/or “support” generally to account for each of the different types of outcome variables. We have reworded the “current study” section of the paper to more specifically state that the goal of scientific messaging strategies has been to increase public support for policies (like legalization). We believe that this connects better to the beginning of the introduction which specifies that the Gateway Belief Model aims to explain how communicating about scientific consensus may indirectly influence change in people’s support for policies (line 43).

---

## [Editor Report · Decision Letter 1]

9 Nov 2021

Influences of Study Design on the Effectiveness of Consensus Messaging: The Case of Medicinal Cannabis

PONE-D-21-22069R1

Dear Dr. Landrum,

We’re pleased to inform you that your manuscript has been judged scientifically suitable for publication and will be formally accepted for publication once it meets all outstanding technical requirements.

Kind regards,

Lucy J Troup, Ph.D

Academic Editor

PLOS ONE

Additional Editor Comments (optional):

Thank you so much for working with the reviewers comments to improve the manuscript. This is an important topic in light of many policy changes occurring around the use of cannabis as medicine.
---

## [Editor Report · Acceptance letter]

15 Nov 2021

PONE-D-21-22069R1 

Influences of Study Design on the Effectiveness of Consensus Messaging: The Case of Medicinal Cannabis 

Dear Dr. Landrum:

I'm pleased to inform you that your manuscript has been deemed suitable for publication in PLOS ONE. Congratulations! Your manuscript is now with our production department. 

Kind regards, 

on behalf of

Dr. Lucy J Troup 

Academic Editor

PLOS ONE